# Management of Filamentous Fungal Keratitis: A Pragmatic Approach

**DOI:** 10.3390/jof8101067

**Published:** 2022-10-11

**Authors:** Jeremy J. Hoffman, Simon Arunga, Abeer H. A. Mohamed Ahmed, Victor H. Hu, Matthew J. Burton

**Affiliations:** 1International Centre for Eye Health, London School of Hygiene and Tropical Medicine, London WC1E 7HT, UK; 2Sagarmatha Choudhary Eye Hospital, Lahan 56500, Nepal; 3Department of Ophthalmology, Mbarara University of Science and Technology, Mbarara P.O. Box 1410, Uganda; 4National Institute for Health Research Biomedical Research Centre for Ophthalmology at Moorfields Eye Hospital NHS Foundation Trust and UCL Institute of Ophthalmology, London EC1V 9EL, UK

**Keywords:** microbial keratitis, fungal keratitis, management, antifungals, microbiology, natamycin, chlorhexidine

## Abstract

Filamentous fungal infections of the cornea known as filamentous fungal keratitis (FK) are challenging to treat. Topical natamycin 5% is usually first-line treatment following the results of several landmark clinical trials. However, even when treated intensively, infections may progress to corneal perforation. Current topical antifungals are not always effective and are often unavailable. Alternatives topical therapies to natamycin include voriconazole, chlorhexidine, amphotericin B and econazole. Surgical therapy, typically in the form of therapeutic penetrating keratoplasty, may be required for severe cases or following corneal perforation. Alternative treatment strategies such as intrastromal or intracameral injections of antifungals may be used. However, there is often no clear treatment strategy and the evidence to guide therapy is often lacking. This review describes the different treatment options and their evidence and provides a pragmatic approach to the management of fungal keratitis, particularly for clinicians working in tropical, low-resource settings where fungal keratitis is most prevalent.

## 1. General Management of Fungal Keratitis

Treatment for filamentous fungal keratitis (FK) is usually with topical antifungal agents. Surgical intervention, usually in the form of a corneal transplant or therapeutic penetrating keratoplasty (TPK), is generally reserved for cases of corneal perforation, relentless progression (despite medical treatment) and for visual rehabilitation once the acute infection has resolved and the cornea is scarred. In Africa and other settings where TPK is not readily available, there is a limited role for temporary tarsorrhaphy and conjunctival flap to offer tectonic support in advanced cases at risk of perforation. There are a limited number of antifungals available with action against fungal keratitis, of which there are four main groups: imidazoles, triazoles, polyenes and fluorinated pyrimidines. These may be available topically, orally or by intravenous injection. Subconjunctival injection or corneal stromal injection may also be given.

## 2. Medical Management of FK

### 2.1. Topical Treatment

Most antifungal agents used for fungal keratitis belong to either the polyene (natamycin and amphotericin) or azole (voriconazole (VCZ), ketoconazole (KCZ), fluconazole, itraconazole, miconazole, and poscaconazole). Polyenes work by binding to ergosterol (essential for maintaining cellular membrane integrity) and inhibiting its cellular functions; additionally, amphotericin B permeabilises the fungal membrane (natamycin does not) [1,2,3]. Azoles, on the other hand, act by inhibiting the biosynthesis of ergosterol [4].

There have been several clinical trials comparing various treatment options for filamentous fungal keratitis over the last few decades, which have been reviewed systematically [5,6]. All trials to date have been conducted in South Asian countries with a high incidence of fungal keratitis.

### 2.2. Natamycin, Voriconazole and Econazole

Natamycin was approved in the 1960s by the FDA and remains the only licensed drug in the USA for FK. It is produced naturally by the bacteria *Streptomyces natalensis* [7]. When manufactured, it is produced as a preserved suspension formulation at a 5% concentration (50 mg/mL). It must be shaken well before application, as it is a suspension. It is active against a wide range of fungal organisms, including *Fusarium*, *Aspergillus*, *Alternaria*, *Candida*, *Cephalosporium*, *Colletotrichum*, *Curvularia*, *Lasiodiplodia*, *Scedosporium*, *Trichophyton*, and *Penicillium* [7,8]. Patients are usually treated initially with one drop hourly (day and night for 48 h and then daytime only), typically for a week, with the frequency reduced depending on the clinical response. Once the infection has resolved, it is advisable to continue with natamycin four times daily for four more weeks [9].

Natamycin has been compared to several other topical agents including voriconazole and econazole [10,11,12,13,14,15]. Voriconazole, a triazole antifungal agent licensed for the treatment of invasive aspergillosis, is used off-label as a topical formulation for the treatment of fungal keratitis. It is not currently routinely available as a pre-prepared eyedrop, and instead must be reconstituted from the powder for injection to achieve a concentration of 10 mg/mL [16]. Once reconstituted, it is advisable to refrigerate it and use within 48 h [16]. Its antifungal activity and dosing regimen is similar to natamycin [17]. An initial prospective randomised controlled trial (RCT) showed no significant difference (*p* = 0.837) between natamycin and voriconazole in terms of primary outcome measure (time to healing of epithelial defect). The authors therefore concluded that voriconazole was “an effective and well-tolerated drug” and larger trials were warranted to demonstrate superiority [10]. Meanwhile, Prajna et al. also compared topical natamycin to voriconazole in a therapeutic exploratory randomised clinical trial; 120 patients were randomised to either natamycin or voriconazole, and some had repeated corneal epithelial scraping. The study also concluded that there was no significant difference between groups for the primary outcome of visual acuity at three months, with a non-significant trend favouring voriconazole (0.98 logMAR better, 95%CI: −0.28 to 0.83, *p* = 0.29). Incidentally, repeated scraping was associated with a worse outcome, although again this was non-significant (*p* = 0.06) [11]. To evaluate the efficacy of voriconazole thoroughly, the Mycotic Ulcer Treatment Trials (MUTT) were developed [12,13]. In MUTT1, topical natamycin 5% was compared to topical voriconazole 1% in a trial that was due to recruit 368 patients but was terminated early on recommendation by the trial Data Safety and Monitoring Committee, as the number of perforations in the voriconazole group were significantly higher than in the natamycin group (34 vs. 18 perforations, *p* = 0.02; after 323 recruited). Vision was −0.18 logMAR better at three months in the natamycin group compared to the voriconazole group (*p* = 0.006) [12]. Sharma et al. also found natamycin to be superior to voriconazole in terms of best spectacle-corrected visual acuity (BSCVA) vision at the final follow-up in a more recent randomised controlled trial that recruited 119 patients (natamycin mean BSCVA LogMAR 0.6 (CI 0.4–0.8), compared to voriconazole mean BSCVA LogMAR 1.1 (CI 0.9–1.2) *p* = 0.01) [14].

There have been two published meta-analysis studies comparing natamycin to voriconazole, with the primary outcome being best spectacle-corrected visual acuity at three months [5,18]. These included the pilot study by Prajna and colleagues (2010), the study by Arora and colleagues, and MUTT1 [11,12,19]. The Cochrane review found a non-significant trend favouring natamycin (Figure 1) [5], whilst McDonald et al. found evidence for natamycin being superior to voriconazole [18]. The discrepancy between these two results is likely due to differences in the inclusion criteria and analysis strategy; the total number of subjects in the analysis by McDonald was 473, compared to 434 in the Cochrane review. The Cochrane review analysis used a more conservative random-effects model rather than a fixed-effect model, as there were only three studies included. There has been no further published peer-reviewed meta-analysis that includes the more recent study by Sharma et al. (2015), although if this study were also to be included, then the pooled estimate of effect would be clearly in favour of natamycin 5% (standardised mean difference 0.34 logMAR, 95% CI 0.17 to 0.50; I^2^ = 73.3%, *p* = 0.011; personal communication from Dr Simon Arunga).

If one considers perforations (or need for TPK) as the outcome measure, there are significantly fewer perforations or TPKs in patients treated with natamycin 5% compared to voriconazole [5,18], as reported by the two meta-analyses (risk ratio favouring natamycin 1.89, 95% CI 1.14 to 3.12; I^2^ = 0%, *p* = 0.01; Figure 2) [18].

Subgroup analysis in MUTT1 demonstrated that the effect of natamycin compared to voriconazole was significantly greater in those infected with *Fusarium* species (40% of patients) compared to those with non-*Fusarium* species (*Aspergillus* 17%, other 43%), for whom there was no evidence of a difference between treatment arms [12]. However, this subgroup analysis was not prespecified in the MUTT1 protocol. Similar findings were reported by the 2015 trial by Sharma [14]. However, it is important to acknowledge that these studies were conducted in South Asian populations, and therefore may not be generalisable to all other settings.

Natamycin has also been compared to econazole 2%, an imidazole antifungal agent, investigating time to healing and proportion healed (“success”) at four weeks [15]. In this trial, 116 patients were recruited in India, of whom the majority were infected with *Fusarium* (55.2%), followed by *Aspergillus* (25.9%), and randomised to either natamycin 5% or econazole 2%. There was no significant difference between the two arms in terms success (log rank 0.52, *p* = 0.47) or for time to heal (epithelial defect, log rank 0.82, *p* = 0.37; infiltrate, log rank 0.86, *p* = 0.35). This study may have been inadequately powered to detect a difference and did not consider vision or perforation rate, unlike the more recent trials, limiting the usefulness of these results. Although the authors suggested that econazole 2% could be an alternative treatment to natamycin, there have been no subsequent randomised controlled trials, likely due to the emergence of voriconazole to the market. A retrospective study by the same group comparing natamycin 5% monotherapy with natamycin 5% plus econazole 2% found no difference between the two arms in terms of “success” (healed or healing ulcer at four weeks). Despite this, it should be considered that econazole is considerably cheaper and more widely available than voriconazole and is often the only topical antifungal available in many Low- and Middle-Income Countries (LMICs) [20].

As with many other eyedrops, when used for a prolonged duration, natamycin can be toxic to the corneal epithelium. Other side effects include burning or stinging when applied, ocular discomfort including foreign body sensation, conjunctival hyperaemia, and epiphora. Topical voriconazole has a similar side effect profile to natamycin, and has additionally been reported to cause periocular dermatitis.

A new soluble form of topical natamycin (natamycin 1% *w*/*v*, “Natasol”) has been developed, with initial experimental animal studies suggesting it is safe with no cases of ocular toxicity reported, as well as being non-inferior to natamycin 5% suspension in terms of its pharmacodynamics and ocular penetration [21]. Further research is warranted, but the authors hope that patient compliance may be increased in this formulation.

Given the strong evidence for natamycin, it is generally considered the first-line agent for fungal keratitis, particularly in areas with a high prevalence of *Fusarium*. For this reason, it has been added to the WHO essential medicine list [22], although its cost outside of South Asia (where it is made as a form of off-patent generic medication) remains prohibitively expensive for many and its availability remains very limited.

### 2.3. Amphotericin B, Fluconazole, and Echinocandins

Although amphotericin B has been used as an alternative, often second-line, agent to treat fungal keratitis, there is very limited evidence with regards to its efficacy other than case reports and case series, some of which have used it in addition to other antifungals [17,23,24]. In its standard form, amphotericin B cannot penetrate the intact cornea. Amphotericin B has been shown to be more effective against yeasts compared to *Fusarium* species [25,26]. There have been no head-to-head trials published to date comparing topical amphotericin B to other topical agents.

Fluconazole is generally considered to be much more effective against yeast infection than filamentous fungal infection. A randomised controlled trial comparing natamycin 5% to fluconazole 0.2% was terminated early after only eight patients were enrolled, as all four patients in the fluconazole arm showed no signs of improvement [27]. However, rather surprisingly, a more recent case series from Paraguay of filamentous fungal keratitis patients treated with either fluconazole 0.2% monotherapy or fluconazole 0.2% and oral ketoconazole 200 mg suggested that fluconazole 0.2% may be effective in filamentous fungal infections with 70% of patients showing resolution of their disease, with no evidence of benefit from the addition of oral ketoconazole [28]. There is currently no convincing evidence to recommend the use of topical fluconazole in filamentous fungal keratitis over the use of natamycin.

The echinocandins are a relatively novel class of antifungal agent that act by blocking β-(1,3)-D-glucan synthesis, which is an important structural component that maintains the fungal cell wall integrity [29], and consists of three agents: caspofungin, micafungin, and anidulafungin. They have been shown to be successful at treating systemic fungal infections [30,31], and in vitro and animal studies have suggested a potential role for their use in fungal keratitis [29,32,33,34,35,36]. However, other than several case reports [29,37,38,39,40,41], there is very limited published evidence regarding their efficacy (particularly as monotherapy), and their use at present is best considered experimental with larger clinical trials warranted.

### 2.4. Chlorhexidine for Fungal Keratitis

Chlorhexidine is an antiseptic agent, with both antibacterial and antifungal properties. It is a widely-used broad-spectrum biocide, killing microorganisms through cell membrane disruption [42,43,44]. For example, chlorhexidine 0.2% *w/v* solution is very widely used as a long-term mouth wash for the prevention and treatment of oral candidiasis (a fungal infection) and for general oral hygiene [45,46,47]. Chlorhexidine 0.2% mouth wash is considered to be locally and systemically safe.

Chlorhexidine has been used in ophthalmology for more than 30 years as an eye-drop preservative, sterilizing contact lenses, pre-operative topical antiseptic and for treating *Acanthamoeba* sp. and fungal keratitis [48,49,50,51,52,53]. It is very important to note that all chlorhexidine solutions used topically in ophthalmic practice are aqueous preparations, i.e., they do not contain any detergents or alcohol.

In a study evaluating potential affordable antifungal treatments for keratitis, chlorhexidine digluconate was compared in vitro with propamidine (Brolene), povidone iodine and polyhexamethylene biguanide (PHMB) [54]. Several concentrations of these agents were tested against a panel of 95 fungal keratitis isolates from Ghana and India. The chlorhexidine 0.2% gave the best results in vitro, with inhibition of 90/95 isolates. The investigators then conducted a pilot case series study in India of chlorhexidine digluconate 0.2% in 11 patients with fungal keratitis (7 non-severe and 4 severe cases). They found that 10/11 cases healed with chlorhexidine 0.2%; one severe case did not respond. The study also included a non-randomised comparison group of eight patients with fungal keratitis (seven non-severe, one severe) who were treated with topical econazole (a frequently used treatment at that time). They reported that seven out of eight responded to econazole; the severe case did not respond to the econazole.

Subsequently two pilot RCTs of chlorhexidine for fungal keratitis were conducted. In the first trial, involving 60 patients conducted in south India, three chlorhexidine gluconate concentrations (0.05%, 0.1%, 0.2% *w*/*v*) were compared to each other and to natamycin 5% [49]. There was evidence suggestive that chlorhexidine 0.2% might be better than natamycin 5% both in terms of the proportion showing a favourable response by 5 days (75% vs. 44%) and cure by 21 days (83% vs. 50%). The chlorhexidine 0.2% performed better than both the 0.05% and the 0.1% concentrations. The chlorhexidine 0.2% *w/v* concentration used in this trial is the same as that used in mouthwash, and is systemically safe for oral mucosal application.

In the second trial, involving 70 patients conducted in Bangladesh, chlorhexidine 0.2% was compared to topical natamycin 2.5% (half standard concentration). There was evidence chlorhexidine 0.2% was associated with a favourable response in more cases than natamycin 2.5% by 5 days (89% vs. 51%; RR = 0.23, 95%CI 0.09–0.63) [50]. By 21 days, 44% of the chlorhexidine treated group were cured compared to 28% of the natamycin group.

Overall, a Cochrane systematic review of treatments for fungal keratitis found a non-significant trend favouring chlorhexidine 0.2% over natamycin 5% in “curing” by 21-days (RR = 0.70, 95%CI 0.45–1.09) when the data from these two trials was combined; see Figure 3 [5].

Chlorhexidine 0.2% has recently been compared to natamycin 5% in a non-inferiority trial conducted in Nepal [55,56]. This trial found strong evidence to suggest that natamycin-treated participants had significantly better 3-month BSCVA than chlorhexidine-treated participants, after adjusting for baseline BSCVA (regression coefficient, −0.30; 95% confidence interval [CI], −0.42 to −0.18; *p* < 0.001). There was no difference in re-culture positivity between arms at day 7. The majority of chlorhexidine-treated patients healed (151/175, 86.3%), although this proportion was smaller than that for natamycin-treated cases (163/173, 94.2%; *p* = 0.018). Natamycin-treated cases were less likely to perforate or require an emergency corneal graft, after adjusting for baseline ulcer depth (odds ratio, 0.34; 95% CI, 0.15–0.79; *p* = 0.013).

Although treatment with natamycin was associated with significantly better visual acuity, with fewer perforations, compared to treatment with chlorhexidine, the proportion of healed chlorhexidine-treated cases is comparable to that of voriconazole reported in earlier trials [11,12,57]. The results from this trial suggest chlorhexidine appears to be effective in the “sterilisation phase” initially, but prolonged treatment may lead to delayed re-epithelialisation and worse scarring. A reduced concentration or reduced dosing frequency following initial sterilisation may lead to improved healing and visual outcome; more research into this area is warranted. These results suggest natamycin remains the preferred first-line monotherapy treatment for filamentous fungal keratitis, whilst chlorhexidine 0.2% may be considered in situations where natamycin is unavailable. A recent case series from Uganda suggests a potential adjunctive role for chlorhexidine 0.2% in addition to natamycin 5% in recalcitrant cases [58], with a large randomised controlled trial currently underway to explore this role further [59].

### 2.5. Oral Treatment

Adjunctive oral treatment for FK, with either itraconazole, ketoconazole or voriconazole, has been investigated, although the added benefit of these oral agents remains uncertain [9]. In MUTT 2, again in conducted in South Asia (India and Nepal), 240 patients with severe fungal keratitis (BSCVA logMAR 1.3 or worse) were randomised to either oral voriconazole or placebo, with all patients receiving topical treatment (initially topical voriconazole monotherapy prior to the results of MUTT 1, then in combination with natamycin, two years into recruitment). There was no difference in primary outcome (perforation rate or corneal graft) within three months between groups (hazard ratio, 0.82; 95% CI, 0.57–1.18; *p* = 0.29), with more side effects reported in the voriconazole group (*p* < 0.001). The study therefore concluded that there was no benefit in adding oral voriconazole in the treatment of severe filamentous fungal corneal infections [13]. A prespecified subgroup analysis suggested that *Fusarium*-infected patients treated with oral voriconazole may have a reduced rate of perforation, although this was not statistically significant.

An earlier RCT of 54 FK patients treated with topical itraconazole compared oral itraconazole to no oral treatment (i.e., no placebo in the control arm) [60]. In terms of healing by six weeks, there was no evidence of a difference between arms (RR 1.0, 95%CI 0.37–2.71).

More recently, an RCT of 50 patients (all treated with topical natamycin) with severe FK were randomised to either oral voriconazole or oral ketoconaczole [61]. There was evidence that patients treated with oral voriconazole had 0.26 LogMAR better BSCVA at three months compared to those treated with oral ketoconazole (95% CI, 0.04–0.48; *p* = 0.02). There was no difference in perforation rates between these two groups (*p* = 0.45).

Oral antifungal therapy has the potential to cause severe, even life-threatening, adverse effects, primarily in the form of hepatotoxicity. Dosing for voriconazole should be adjusted for body weight, with adults with a body weight under 40 kg receiving 200 mg twice a day for two doses (as loading doses) and then 100 mg twice a day for the duration of therapy [62]. Adults 40 kg and above should receive double this dose. Bioavailability of voriconazole is good, with peak plasma concentration achieved between 1.43 and 1.81 h after ingestion, although it should be taken 1 h before or 2 h after eating [63,64]. Baseline and then weekly liver function tests (LFTs) are necessary due to the risk of hepatotoxicity; these can be reduced to monthly if there is no change in LFTs during the initial one month of treatment [65]. In addition, oral voriconazole can cause visual disturbance, change in colour vision, and photophobia. These are typically transient and occur in one third of patients, occurring approximately 30 min after administration and lasting for 30 min. Rarely, visual hallucinations, confusion, and psychosis have been reported [66].

Given the limited evidence for any additional adjunctive benefit and potential for systemic side effects, adjunctive oral voriconazole in FK remains controversial and should not be given routinely, even for severe disease. The need for liver function testing is an additional challenge to many settings where FK is common, as such tests may not be easily accessible or affordable.

### 2.6. Injected Drug Delivery

In addition to topical treatment, injections of antifungals into either the corneal stroma (i.e., intrastromal injection) or anterior chamber (i.e., intracameral injection) have also been performed in severe disease where the response to topical treatment has been inadequate [67,68]. Options that have been reported in the literature include intracameral amphotericin B or voriconazole, and intrastromal amphotericin B or voriconazole [9,66,69]. Natamycin has generally been avoided in a targeted manner as it is usually formulated as a suspension and initial animal studies did not recommend its use [70]; however, there have been recent experimental studies using a new soluble form of the drug (sterile unpreserved Natasol 0.01% intrastromal injection), that suggest it may have a role to play [21,71]; further evaluation is necessary.

Intrastromal injections are performed by injecting a suitable antifungal (e.g., voriconazole 50 μg/0.1 mL) loaded within a 1 mL tuberculin syringe with a 30-gauge needle, which is inserted obliquely into the clear, uninvolved cornea to reach just adjacent to the ulcer within the mid-stromal level [67]. The drug is deposited circumferentially around the ulcer by giving five divided doses, resulting in the drug surrounding the ulcer in each meridian. This can be repeated, with 72 h between injections. There are inherent risks involved, including spreading infection to new foci, intraocular inflammation, cataract formation, perforation, raised intraocular pressure, hyphaema, and damage to the corneal endothelium [9].

Intrastromal voriconazole injections have been compared to topical therapy alone in two randomised controlled trials [72,73]. The first was a randomised controlled trial of 40 patients who were not responding to initial therapy with natamycin 5%. Patients were randomised to either topical voriconazole 1% alone or to intrastromal injections of voriconazole 50 μg/0.1 mL [72]. The authors found that patients receiving topical voriconazole had a mean BSCVA of -0.397 better than the intrastromal injection group (*p* = 0.008). Additionally, 19/20 patients receiving topical voriconazole healed with therapy. The authors concluded that topical, as opposed to intrastromal, voriconazole may be beneficial in addition to natamycin in recalcitrant disease not responding to natamycin 5% monotherapy [72]. The second, more recent, clinical trial compared natamycin 5% monotherapy to natamycin 5% and intrastromal voriconazole in 70 patients with fungal keratitis [73]. The authors found no evidence of a difference between the groups for all outcome measures investigated (BSCVA, scar size, perforation rate, and microbiological cure) and concluded that “studies consistently suggest voriconazole has a limited role in the treatment of filamentous fungal ulcers” [73]. Based on these two studies, there is currently no clinical trial-level evidence to support intrastromal injections of voriconazole. There have, however, been several case series that suggested intrastromal voriconazole might be beneficial in treating deep infiltrates or stromal abscesses that were unresponsive to first- and second-line topical and medical therapy (natamycin 5%, voriconazole 1%, oral itraconazole or ketoconazole) [68,74,75,76]. Therefore, intrastromal voriconazole injections may have a limited role in select cases refractory to initial medical therapy.

There are very limited published data on the use of intracameral voriconazole. One case series evaluated five fungal keratitis patients with endoexudates who were treated with a single dose of intracameral voriconazole 50 mcg/0.1 mL, followed by voriconazole 1% topical therapy [77]. All cases demonstrated a complete resolution of infection within 3 weeks to 3 months. Another case series assessed intracameral voriconazole amongst 10 patients with fungal endophthalmitis resulting from keratitis in China [78]. This used 100 mcg/0.1 mL intracameral voriconazole, and was repeated up to eight times over the course of the disease. The authors reported that “all cases of the fungal anterior chamber invasion resolved after treatment” [78], and four eyes required therapeutic penetrating keratoplasties. Therefore, there may be a limited role for intracameral voriconazole injection in patients with very deep fungal keratitis with anterior chamber spread of the fungus.

There is emerging evidence for the use of intracameral amphotericin B (ICAB) for patients with deep and/or recalcitrant filamentous fungal infection. There have been three published small randomised controlled trials evaluating this treatment. In the first trial of 42 patients, conducted in China, patients received topical and oral fluconazole and either intracameral amphotericin B or a sham injection, with those receiving intracameral amphotericin B exhibiting a faster healing time (*p* = 0.001) [79]. However, in the second trial, an ICAB (5 µg in 0.1 mL 5% dextrose) injection, in addition to topical natamycin 5% and oral ketoconazole, was evaluated in 45 patients with deep fungal keratitis [80]. There were three arms in this trial: medical therapy alone; medical therapy and intracameral amphotericin B; and medical therapy, intracameral amphotericin B and anterior chamber washout. This trial found no difference between the arms in terms of treatment success (primary outcome measure), time to disappearance of hypopyon, time to healing, final visual acuity, or healing. However, this study may have used too small a sample to detect a difference, with only 15 patients in each arm. In contrast, two other non-randomised prospective interventional studies showed improved outcomes for patients treated with ICAB [81,82]. Non-responding (to medical therapy) patients treated with ICAB had a significantly greater mean improvement in BSCVA (*p* < 0.01) compared to responding patients not treated with ICAB, faster clearance of hypopyon (*p* < 0.01), fewer scars and complications (*p* < 0.05) [82]. In the other study, all patients were given ICAB in addition to medical therapy, with one arm receiving it early (2 weeks) in contrast to the other arm receiving it late (4 weeks). Healing time was significantly faster in the early group (*p* < 0.001) compared to the late group, whilst there were no perforations (0/25) in the early group, compared to 10/25 in the late group (*p* = 0.006) [81]. Several case series also suggest good outcomes for patients treated with intracameral amphotericin B [83,84,85,86]. ICAB may therefore be an appropriate treatment in select cases including recalcitrant deep filamentous fungal keratitis cases with hypopyon or anterior chamber involvement. An adequately powered randomised controlled trial would be helpful to answer this more clearly.

Interestingly there is only limited published evidence on the use of intrastromal amphotericin B injection on its own, as several case series include it in combination with another intrastromal antifungal (either voriconazole or fluconazole). One audit from Egypt that retrospectively reviewed cases of FK not responding to topical therapy found that those treated with a single intrastromal injection of fluconazole and amphotericin B healed faster than those who received amphotericin B monotherapy [87]. A case series of 32 patients with progressive FK despite 10 days of combined topical voriconazole 1% and amphotericin B 0.15% found that 87.5% of patients demonstrated complete resolution of their fungal keratitis, with the remainder requiring TPK [88]. Patients received between 1 and 18 intrastromal injections (mean 9.3 ± 6.4).

Intrastromal amphotericin B (5 mcg/0.1 mL), intrastromal voriconazole (50 mcg/0.1 mL), and intrastromal natamycin (10 mcg/0.1 mL) injections have recently been compared in a three-way randomised controlled clinical trial in fungal keratitis patients not responding to two weeks of topical natamycin 5% therapy (sixty eyes in total, with twenty in each group) [89]. Included patients had more than 50% stromal involvement and an ulcer size of more than 2 mm. This found the mean duration of healing to be faster in the intrastromal natamycin group compared to the other two groups (*p* = 0.02). There was no evidence of a difference between the groups in terms of healing success, although there was significantly more deep vascularisation in the intrastromal amphotericin arm.

Overall evidence is lacking to make clear recommendations for the routine use of either intrastromal or intracameral injections of antifungals, although these techniques can be considered in select (typically deep-seated) cases not responding to topical treatment.

## 3. Surgical Treatment

### 3.1. Therapeutic Penetrating Keratoplasty

Surgical management, typically in the form of TPK, is an important step in managing a significant proportion of patients with fungal keratitis, predominantly for patients with perforation, impending perforation, and cases not improving despite maximal medical therapy [90]. The aim of surgery is to maintain globe integrity, whilst reducing the infectious burden. The percentage of patients requiring TPK is somewhat variable, with reports in the literature suggesting between 15–55% of FK patients required a TPK during the course of their disease [13,90,91,92,93,94]. In MUTT2, the risk factors for requiring TPK or developing a corneal perforation include presence of hypopyon (OR 2.28, 95% CI 1.18–4.40, *p* = 0.01), increased infiltrate size (OR 1.37, 95% CI 1.12–1.67, *p* = 0.02, for each additional 1 mm in infiltrate size), and increasing infiltrate depth (for each step increase in infiltrate depth (no infiltrate, anterior one-third, up to two-thirds stromal depth, or involving posterior one-third of stroma) OR 1.69, 95% CI 1.12–2.53, *p* = 0.01) [95].

Unfortunately, graft survival is often poor for patients with FK due to the active infection and inflammation, which increases the likelihood of graft rejection, secondary or re-infection, and secondary glaucoma [90]. Recurrence within the graft has been reported from 0–47% [92,96,97,98,99,100], and secondary glaucoma in 2–64% of cases [90,92,93,96,101].

TPK success can be considered in terms of anatomical integrity, graft clarity, and visual improvement. These have been considered by several retrospective studies. Anatomical integrity has been reported to range from 64–97% [90,96,97,99,102,103], graft clarity from 26–94% [92,96,97,98,99,100], and visual improvement in 6–88% of cases [93,102]. However, these studies did not specify whether secondary optical keratoplasties were performed.

TPK has a crucial role to play in FK by saving the eye and controlling the infection. Unfortunately, however, there is a lack of donor cornea material in many LMICs where the need is greatest. Where tissue is available, the quality is often poor, with low endothelial cell counts [104]. It has also been suggested that outcomes of TPK for fungal infection in LMICs may be poor due to delayed use of topical steroids post-operatively due to the fear of re-infection, meaning inflammation, graft decompensation, and vascularisation are more commonly encountered [90,104].

### 3.2. Lamellar Keratoplasty

Lamellar keratoplasty (LK) as an alternative surgical intervention to treat fungal keratoplasty has been reported in the literature [105,106,107,108,109]. Unlike TPK, which is most frequently performed for perforations (either impending or frank), in the context of FK, lamellar keratoplasty is usually performed to surgically resect infections not extending into the anterior chamber. LK has an advantage over TPK in that corneal donor material can be preserved in glycerol or dehydration [110,111], meaning that supplying graft material for LMICs where there is a shortage of fresh corneal donors is potentially possible, although there remain considerable logistical and legal challenges to overcome before this can be truly viable. There have also been some published case series of using acellular porcine donor corneas, with promising results including low recurrence rates and good visual recovery [112,113]. By preserving the host endothelium, the risk of endothelial immune rejection post-operatively is removed [114]. However, LK is a technically more challenging procedure than TPK and surgeons will still need to convert to a TPK in the event of perforation, meaning that the advantage of using preserved donor material is lost.

LK has been compared to TPK in two retrospective studies for fungal keratitis [105,115]. An early study from India in the 1960s concluded that “lamellar keratoplasty invariably fails, because the fungus is able to penetrate Descemet’s membrane” after it found that 6/7 LK cases developed re-infection, compared to only 1/10 cases in the TPK group [115]. However, a recent retrospective study of 94 LK cases and 161 TPK cases from China using modern surgical techniques found no difference in recurrence rates between groups in terms of recurrence, whilst the immune rejection rate was significantly lower in the LK group (1.1% vs. 18.6%, *p* < 0.001), along with secondary glaucoma (*p* = 0.018). There was no difference in visual acuity or refractive outcome between groups [105]. Another comparative study conducted for infective keratitis cases (including non-fungal cases) found no significant difference between LK and TPK in terms of therapeutic success (*p* = 0.74), whilst LK patients had a significantly greater mean improvement in visual acuity compared to TPK patients (7.27 lines versus 4.76 lines, *p* = 0.01) [116].

Whilst these results suggest a potential role for LK in the surgical management of superficial medically recalcitrant FK, large-scale prospective studies are needed to evaluate this more definitively. LK may prove useful in settings where human donor corneal material for full-thickness penetrating keratoplasty is less readily available, potentially with further development of acellular porcine donor corneas, but this needs to be balanced against the more technically challenging nature of the procedure.

### 3.3. Amniotic Membrane Grafts and Conjunctival Flaps

The amniotic membrane is the innermost layer of the placenta, consisting of a single layer of epithelium, a thick basement membrane, and an avascular stromal matrix [117]. It exhibits a diverse range of beneficial biological properties, including anti-inflammatory and antimicrobial, as well as promoting wound healing [118]. Unlike corneal tissue, amniotic membrane donor tissue is widely available, easy to store, and not prone to graft rejection. It has been used to treat a wide range of ocular surface disorders including microbial keratitis, chemical eye injury, corneal perforation, and limbal stem cell deficiency, amongst others [118,119,120].

In managing microbial keratitis, amniotic membrane grafts (AMG) have typically been used as second-line therapy to promote corneal healing in cases of persistent epithelial defect following the sterilisation phase. However, there is emerging evidence from a recent systematic review and meta-analysis that there is a benefit in early adjuvant AMG in terms of more rapid corneal healing and improved visual outcome for moderate-to-severe fungal keratitis [120]. This review highlighted two RCTs and one non-randomised controlled study that compared AMG to standard antimicrobial treatment [121,122,123], and calculated the pooled estimate of the time to complete corneal healing to be 6.90 days faster in the AMG group (mean difference − 6.90 days; 95% CI − 11.58 to − 2.21; *p* = 0.004) [120].

Conjunctival flaps are often used in LMICs, particularly in the African region for recalcitrant fungal keratitis, as it is a relatively straightforward, low-cost procedure and does not require any donor tissue. It is believed that healing is promoted by placing a vascular bed over the ulcer, as well as controlling infection and protecting against small perforations [124,125]. Despite conjunctival flaps often being used, there is limited published work regarding their efficacy. There has been one randomised controlled trial that compared AMG to conjunctival flaps and found no difference between arms in terms of re-epithelialisation time, persistence of infection, complications and visual improvement [126]. Conjunctival flaps have also been used in addition to AMG. One case series from China of 17 patients with FK refractory to medical treatment who underwent conjunctival flaps in addition to AMG found that the globe was preserved in 15/17 (88%) of cases, whilst there were no cases of raised intraocular pressure [127]. Conjunctival flaps may therefore be an appropriate treatment – either alone or in addition to AMG – for severe recalcitrant FK without large perforations.

### 3.4. Corneal Collagen Cross-Linking

Corneal collagen cross-linking (CXL) is a well-established technique that is typically used in preventing progression of corneal ectasias such as keratoconus and pellucid marginal degeneration. In brief, CXL involves ultraviolet-A irradiation of the cornea primed with riboflavin (vitamin B2, a photosensitiser), resulting in the formation of oxygen free radicals which then form covalent bonds between the stromal collagen fibrils, increasing the corneal biomechanical stability. Recently, CXL has been considered for the treatment of FK [128]. There are several claimed mechanisms of action, including:Direct anti-microbial action by damaging the pathogens’ DNA/RNA [129]Increasing the resistance to the microorganisms’ enzymatic destruction of the corneal stroma [130]Enhanced corneal penetration of antifungals [131]

However, the evidence for this is limited, with heterogenous protocols and conflicting results. There have been four RCTs to date comparing CXL and medical treatment to medical treatment alone. The first study from India of deep, recalcitrant FK cases was ended early after significantly more patients in the CXL perforated than in the medical arm; [132] the numbers recruited were therefore low. A second study from China showed better results for the CXL arm, but this was also a small trial of only 41 patients and the baseline severity of the cases was not reported [131]. The CLAIR trial, a larger, more recent RCT from India that randomised 111 patients with FK, found that patients who underwent CXL plus topical therapy had significantly higher culture re-positivity at 24 h and worse vision at 3 months, regardless of whether they received natamycin or amphotericin B as topical therapy [130]. A subsequent analysis concluded that the reason for the worse vision in the CXL group was corneal scarring and astigmatism [133]. These findings are in contrast to another recent trial from India of superficial cases, that found patients treated with CXL healed faster and had better final vision. However, the study was small and unmasked and the analysis was unadjusted for baseline characteristics [134]. These trials have been included in a systematic review and meta-analysis investigating the role of CXL in infectious keratitis, with a subgroup analysis specifically for fungal keratitis performed following the publication of the CLAIR trial [135,136]. This found that CXL did not confer any additional benefit or harm in terms of infiltrate size or adverse events, although an unequivocal recommendation regarding CXL use could not be made due to the insufficient number of trials and uneven covariate distribution. Taken together, these studies suggest that there is no role for CXL in severe, deep-seated FK due to the increased risk of perforation, whilst there is limited evidence for any benefit in superficial cases.

### 3.5. Argon Laser for Fungal Keratitis

There have been a few case reports on the use of argon lasers as adjunctive treatment for refractory fungal keratitis unresponsive to topical and systemic therapy [137]. Argon laser irradiation is performed using argon blue-green wavelengths, a spot size of 500 μm, pulse duration of 0.10 s, and power ranging from 500 to 900 mW until blanching is observed of the stroma, together with small cavitations that reach the mid-stroma [137]. It is postulated that argon laser may have a similar mechanism of action to corneal epithelial debridement, enhancing the penetration of antifungals [137]. In addition, it may be fungicidal as a result of the thermal effect on the infected tissue [137], as the temperature rises beyond 90 °C when treated [138].

There have been two small RCTs conducted that included argon laser as a treatment arm: one that compared argon laser to intrastromal voriconazole injection [139] and the other comparing argon laser to AMG [140]. Both included patients that were not responding to topical therapy by day 7. Both trials showed a significantly faster healing time with the argon laser. There was no statistically significant evidence of a difference in vision between the two groups [139,140].

Whilst these studies show some promise for the role of argon lasers, larger, more robust RCTs are warranted to investigate this further before the use of argon lasers can be recommended routinely for treatment-refractory cases of FK; at present it remains experimental.

## 4. Management Strategy: A Pragmatic Approach

Developing standardised guidelines applicable to all settings for the management of FK is challenging, due in part to differences in what treatment options and facilities are available, the prevailing fungal organisms, and epidemiological factors. Effectively evaluating any management strategy is also difficult. In spite of this, the so-called TST Protocol (Topical, Systemic and Targeted Therapy) has been developed based on the evidence outlined above, with the authors reporting the outcomes after four years of implementation on 223 cases [76]. This protocol recommends initial monotherapy with natamycin 5% hourly for 48 h, then every two hours during waking hours until complete re-epithelialisation. At this point, the dose is reduced to approximately four times a day for a further three weeks. Cycloplegia and pain relief are given. Patients with deep (>50% stromal depth), large (>5 mm) ulcers are prescribed a systemic antifungal in addition (oral KCZ administered 200 mg administered twice daily with meals, or oral VCZ 200 mg twice daily 2 h after a meal). If there is poor response by day 7–10 after starting treatment, then topical voriconazole 1% is added, following the same dosing regimen as natamycin. If patients were still responding poorly after a further 7–10 days, then intrastromal and/or intracameral injections of antifungals were performed. These are repeated up to a maximum for four injections, 72 h apart. Patients still not responding despite targeted therapy, patients with significant corneal thinning contraindicating intrastromal injections, or those with frank corneal perforations undergo TPK. The treatment “success” rate in this group (healing without perforation or requiring TPK) was 79.8% overall, i.e., 20.2% of patients required TPK. This compares to 16% and 43.8% in MUTT 1 and 2, respectively [12,13]. Treatment success was 89% in patients receiving intrastromal voriconazole, compared to 63.1% in the medical management group. There was no comparative arm to this study, limiting conclusions, but it does highlight a potential rational, step-wise approach to managing FK.

In view of the above evidence, first-line management of filamentous FK is usually with topical natamycin 5% when it is available. It was added to the WHO Essential Medicines List in 2017 for this indication. However, even when intensive topical natamycin is initiated, infections frequently progress relentlessly to perforation and loss of the eye in ~25% of cases, Figure 4 [12,13,20].

Additional and alternative drugs are clearly needed if the outcome of these infections is to improve. Moreover, in many countries antifungal eye drop treatments are simply not available. This includes most countries in Africa, some Asian countries and some countries in Europe. Natamycin is relatively expensive even if it is available.

Chlorhexidine 0.2% has been considered as a potential alternative agent that could be readily available [55]. However, in a recent large randomised controlled trial conducted in Nepal, natamycin has been shown to be superior to chlorhexidine for the treatment of filamentous FK and remains the first-line treatment [56]. Whilst every effort should be made to ensure that natamycin is widely available and affordable to those most in need, in the meantime chlorhexidine can be considered an alternative, “better than nothing” agent. Treatment algorithms that recommend differing strategies (topical, systemic or targeted therapy) depending on the size and depth of the corneal ulcer at baseline, and how it responds to initial therapy, may result in improved outcomes and may explain our relatively low “failure” rate overall. Natamycin should be the first-line agent in such a protocol, with chlorhexidine substituted should natamycin not be available (Figure 5). Chlorhexidine may have a role as an adjunctive agent to natamycin [58], with work currently under way in East Africa to investigate this [59]. Although large-trial data are lacking, topical amphotericin B could be considered when other agents are unavailable, or as an adjunctive agent in challenging cases. Further clinical trials are required to investigate this potential role further.

## 5. Conclusions

Fungal keratitis is challenging to treat. The current first-line treatment for filamentous fungal keratitis is with topical natamycin 5% monotherapy, supported by a robust evidence base. Additional agents can be used either topically, as targeted injections, or systemically in more challenging cases, but evidence for their benefit is less established. There remains a clear role for surgical intervention, usually in the form of therapeutic penetrating keratoplasty in eyes that have perforated; earlier surgical intervention may be helpful, but requires further research before clear recommendations can be made.

Unfortunately, there remain significant challenges for the management of fungal keratitis, largely attributed to the fact that most patients who suffer with this blinding condition are some of the poorest in the world, frequently neglected and marginalised. Early antifungal treatment is hampered by poor access to services and inappropriate treatment with traditional medicine and steroids. Compounding this, natamycin is frequently unavailable, or if it is, is unaffordable. Alternative agents are therefore warranted, with a clear need for well-designed clinical trials to provide definitive evidence for their use.

## Figures and Tables

**Figure 1 jof-08-01067-f001:**
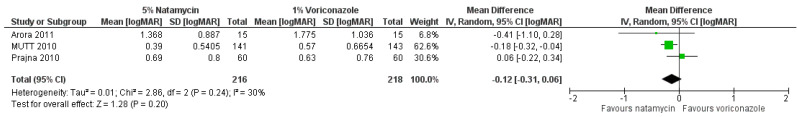
Forest plot of topical natamycin 5% versus voriconazole 1% (outcome: best spectacle corrected visual acuity, logMAR). Reproduced from FlorCruz NV, Evans JR. Medical interventions for fungal keratitis. The Cochrane Database of Systematic Reviews 2015; **4**: CD004241 with permission from John Wiley and Sons [5].

**Figure 2 jof-08-01067-f002:**
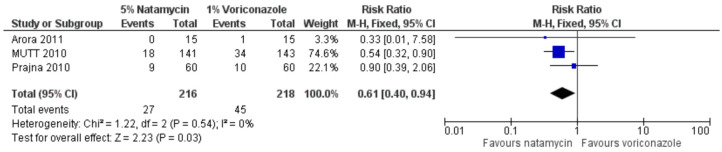
Forest plot of topical natamycin 5% versus voriconazole 1% (outcome: corneal perforation). Reproduced from FlorCruz NV, Evans JR. Medical interventions for fungal keratitis. The Cochrane Database of Systematic Reviews 2015; **4**: CD004241 with permission from John Wiley and Sons [5].

**Figure 3 jof-08-01067-f003:**
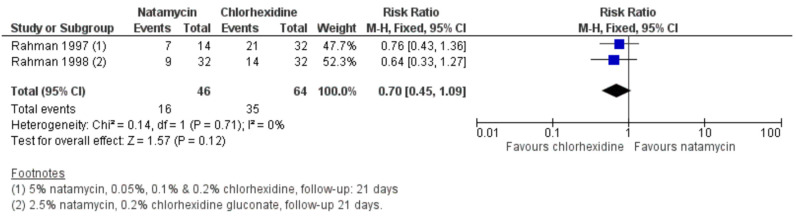
Forest plot of topical natamycin 5% versus chlorhexidine 0.2% (outcome: clinical cure). Reproduced from FlorCruz NV, Evans JR. Medical interventions for fungal keratitis. The Cochrane Database of Systematic Reviews 2015; **4**: CD004241 with permission from John Wiley and Sons [5].

**Figure 4 jof-08-01067-f004:**
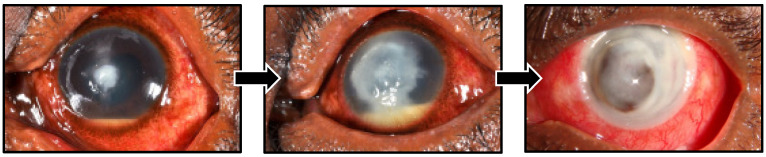
Progression of Fusarium fungal keratitis in a Tanzanian patient despite prompt treatment with topical natamycin 5%. First photograph, baseline presentation; second photograph, progression of infiltrate and increasing hypopyon at one week despite admission and intensive natamycin 5% treatment; third photograph, corneal perforation at three weeks following presentation.

**Figure 5 jof-08-01067-f005:**
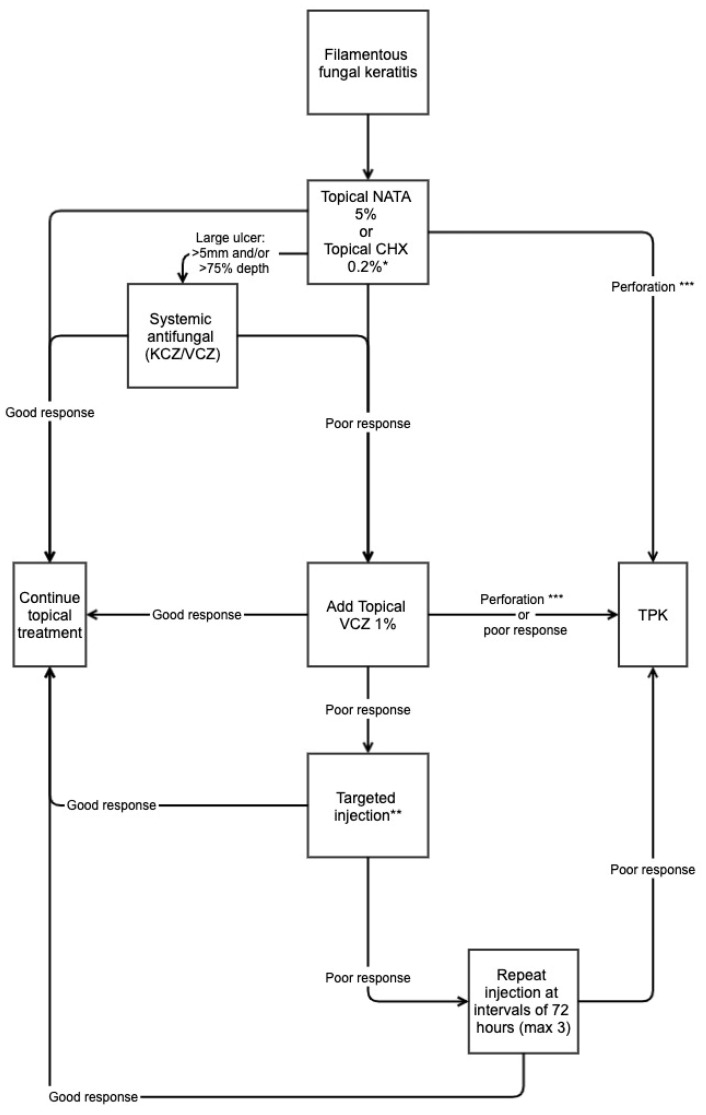
Suggested treatment protocol for filamentous fungal keratitis. * Topical Natamycin 5% is first-line treatment but chlorhexidine 0.2% can be given if natamycin is unavailable. ** Targeted injection refers to either intrastromal/intracameral voriconazole/amphotericin B. *** Perforation includes descemetocele, impending, and frank corneal perforations. Adapted from the TST Protocol [76].

## Data Availability

Not applicable.

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
