# Peer review of "Management of Filamentous Fungal Keratitis: A Pragmatic Approach"

_jof, 2022, doi:10.3390/jof8101067_

Round 1

Reviewer 1 Report

This review manuscript is very well written and resourceful. The authors summarize the current status of fungal keratitis and point directly at the problem of drug choices in LMICs. The pragmatic approach is clear and reasonable. This review should be accepted with minor revisions.

1.       Name of authors (should not include FRCOphth, PhD). The citation (on the left, page 1) has to be corrected as well.

2.       Reference format. The reference list is inconsistent. Mistakes at capitalization of each word in the article title. Name of organism has to be italics, for example.

One question of ref .8. Is this citation proper? The text mentioned several Genera but the reference reported the study on Fusarium and Aspergillus.

3.       If possible, should have include the list of abbreviations.

4.       The full stop should be placed after reference bracket (in text citation).

Author Response

Dear Reviewer

Thank you very much for taking the time to review our manuscript. Your comments have helped to improve the article substantially.

Please see our responses to your questions below:

This review manuscript is very well written and resourceful. The authors summarize the current status of fungal keratitis and point directly at the problem of drug choices in LMICs. The pragmatic approach is clear and reasonable. This review should be accepted with minor revisions.

Thank you for this comment.

  1. Name of authors (should not include FRCOphth, PhD). The citation (on the left, page 1) has to be corrected as well.

This has been amended as requested. Thank you.

2. Reference format. The reference list is inconsistent. Mistakes at capitalization of each word in the article title. Name of organism has to be italics, for example.

Thank you for this comment. I have updated the references as requested.

One question of ref .8. Is this citation proper? The text mentioned several Genera but the reference reported the study on Fusarium and Aspergillus.

Thank you for this comment. I have checked and added an additional citation that includes this.

  1. If possible, should have include the list of abbreviations.

Thank you, I agree but unfortunately I don't think this is possible within the context of this journal.

  1. The full stop should be placed after reference bracket (in text citation).

Thank you for this comment, I have updated the formatting accordingly.

Best wishes

Jeremy

Reviewer 2 Report

The manuscript entitled "Management of filamentous fungal keratitis: a pragmatic approach" is an interesting contribution to the field. I found neither major nor minor remarkable revisions; thus I directly recommend acceptance.

Author Response

Dear Reviewer

Thank you for taking the time to review our manuscript and for recommending it for publication.

Kind regards

Jeremy Hoffman

Reviewer 3 Report

The work is well written and addresses an issue of great importance, especially with the increasing number of cases of fungal keratitis in immunocompetent patients.

This increase reflects the adaptive capacity of opportunistic fungi that have  emerged as pathogenic fungi and draws attention to the problem of resistance to antifungals widely used in the clinic.

I congratulate the authors for choosing the topic and recommend the article for publication.

Author Response

(The authors gave the same response as above.)
